# Design and Development of a Wideband Planar Yagi Antenna Using Tightly Coupled Directive Element

**DOI:** 10.3390/mi11110975

**Published:** 2020-10-30

**Authors:** Muhammad A. Ashraf, Khalid Jamil, Ahmed Telba, Mohammed A. Alzabidi, Abdel Razik Sebak

**Affiliations:** 1Department of Electrical Engineering, King Saud University, Riyadh 11421, Saudi Arabia; m.ahmad.ashraf@gmail.com (M.A.A.); khalid.jamil@gmail.com (K.J.); 2Prince Sultan Defense Studies and Research Center (PSDSARC), Riyadh 13322, Saudi Arabia; mohammed.alzabidi@psdsarc.org.sa; 3Department of Computer and Electrical Engineering, Concordia University, Montreal, QC H3G 1M8, Canada; abdo@ece.concordia.ca

**Keywords:** dipole arms, elliptic directive element, end-fire, printed Yagi antenna

## Abstract

In this paper, a novel concept on the design of a broadband printed Yagi antenna for S-band wireless communication applications is presented. The proposed antenna exhibits a wide bandwidth (more than 48% fractional bandwidth) operating in the frequency range 2.6 GHz–4.3 GHz. This is achieved by employing an elliptically shaped coupled-directive element, which is wider compared with other elements. Compared with the conventional printed Yagi design, the tightly coupled directive element is placed very close (0.019λ to 0.0299λ) to the microstrip-fed dipole arms. The gain performance is enhanced by placing four additional elliptically shaped directive elements towards the electromagnetic field’s direction of propagation. The overall size of the proposed antenna is 60 mm × 140 mm × 1.6 mm. The proposed antenna is fabricated and its characteristics, such as reflection coefficient, radiation pattern, and gain, are compared with simulation results. Excellent agreement between measured and simulation results is observed.

## 1. Introduction

In many wireless communication and radar applications, such as TV, microwave imaging, wireless ad-hoc networks, point-to-point and multipoint links, digital array radars, and high-resolution direction tracking, radio frequency (RF) front-end systems are required to operate over a wide bandwidth [1,2,3,4]. For an efficient RF front-end operation, antenna elements offering low cost, light weight, moderate gain, easy integration, and low dispersion to wideband signals are extremely important [5]. To fulfill such requirements, many types of microstrip-based antennas have been proposed in the literature [6,7,8,9,10].

Commonly used broadband microstrip antennas include antipodal and slot-coupled Vivaldi antennas; log-periodic arrays; feedline–coupled, multilayer-stacked patch antennas; and quasi-Yagi antennas [11,12,13,14,15,16]. Generally, Vivaldi and log-periodic array antennas are capable of operating over an ultra-broad bandwidth, but they may not be suitable for some applications where signal bandwidth is smaller. Alternatively, multilayer-stacked patch antennas offer broad bandwidth and can be easily designed and tuned for different operating frequencies by modifying their dimensions. However, the design and implementation of multilayered antennas is complex and less cost-effective [14].

Due to their simple structure, easy integration with other microwave circuits and high gain, Yagi antennas are also suitable in wireless applications. However, typical microstrip Yagi antennas offer a narrow bandwidth (<10%) [17,18]. Research efforts on improving the performance of typical Yagi antennas by exploiting their fundamental design principles have been presented in the literature. As a result, a new antenna structure called quasi-Yagi was proposed to improve the conventional Yagi antenna performance [19]. This antenna was first introduced by Huang in 1991. However, one critical limitation of the Quasi-Yagi antenna is its lower gain compared to the classic Yagi antennas. Several techniques for simultaneously enhancing the bandwidth and gain of microstrip Yagi antennas have been proposed in the literature [20,21,22,23,24,25].

The printed Yagi antenna consists of a dipole antenna acting as a feed, which is called driven element. The directive elements are printed toward the end-fire direction to enhance the gain of the antenna by directing the electromagnetic energy toward a specific direction. The front-to-back ratio and gain over a broad bandwidth or multiple bands can be improved by increasing the number of directive elements. However, this technique increases the overall size of the antenna. For example, a printed Yagi antenna comprising two different types of feeding networks (i.e., loop and dipole types) has been proposed in [26]. This antenna exhibits a fractional bandwidth of 23% and 16% with a gain of 4.7 and 3.65 dBi, respectively. However, the gain enhancement achieved by increasing the number of directive elements is limited as the multiple incorporated directive elements introduce a phase shift at the center frequency of operation in a particular band. Such degradation in antenna performance is not suitable for broadband applications [27]. A broadband Yagi antenna comprising modified bowtie driven dipole is presented in [28]. The presented antenna is very compact with an excellent fractional bandwidth of 42%. However, the maximum gain value is about 4 dBi as it has employed only one directive element to maintain the antenna’s compactness. In [29] a magnetic dipole quasi-Yagi antenna based on a dielectric resonator (DR) is presented. The gain of DR based Yagi antenna is found greater than the traditional ones. However, because of DR high quality factor property, the fraction bandwidth is limited to 3% only. A compact planar Moxon-Yagi composite antenna is presented in [30]. This hybrid antenna operate over dual bands with fractional bandwidth and gain values of 3.2% and 5 dBi, respectively. Similarly, a compact printed Yagi antenna based on half bow-tie monopole element, two director strips and finite ground plane is presented in [31]. The antenna operates at 2.31 GHz center frequency with an excellent fractional bandwidth of 63.5%. As only two directive elements are used, the gain performance of the antenna is compromised to 5.8 dBi. In Table 1, we have compared the performance of our proposed antenna with those published in [28,29,30,31]. It can be inferred that in printed Yagi antenna designs, simultaneous improvement in gain, bandwidth and size is quiet uncommon.

In this work, a moderate-size modified printed Yagi antenna is presented. This antenna is designed to exhibit high gain and wide bandwidth, simultaneously. The conventional printed Yagi antenna comprises a driven element followed by multiple directive elements of a typical rectangular shape. As the electromagnetic energy is coupled from the driven element to the directive element, the inter-element spacing must be in the range 0.15λ–0.3λ. Moreover, the typical rectangular shape can affect matching over a wide bandwidth due to rapid changes in the input impedance. Therefore, in the proposed printed Yagi antenna design, an elliptical shape for the dipole antenna and directive elements was selected. Additionally, contrary to the typical spacing (0.15λ to 0.3λ) used for the directive elements, the input impedance can be matched over a wider bandwidth by placing the first directive element in a closer proximity (i.e., between 0.019λ to 0.0299λ). Thus, the proposed antenna exhibits a wide bandwidth (49.3% fractional bandwidth) covering the S-band from 2.6 GHz to 4.3 GHz. In the proposed design, the directive elements are printed on both the top and bottom sides of the substrate. The bottom-plane directive elements are placed by some offset with respect to the top-plane directive elements. This enhances the effective length toward the direction of propagation, resulting in enhanced gain without increasing the overall dimensions of the antenna.

## 2. Printed Yagi Antenna Design Principles

Antenna design based on elliptical shaped microstrip structures has been emerged as unique and important category among the different shaped because of its dual resonant feature and flexibility in optimization. Design of an elliptical shape microstrip patch antenna has more degree of freedom as the resonant frequency is the function of its major axis and minor axis radii as following [32]:(1)$$f_{11}^{e,o}=\frac{15}{\pi ea}\sqrt{\frac{q_{11}^{e,o}}{\epsilon_{r}}}$$
where, $f_{11}^{e,o}$ is the dual resonant frequency (in GHz) of $TM_{11}^{e}$ and $TM_{11}^{o}$ modes with a and e correspond to physical semi major axis (in cm) and eccentricities, respectively. The eccentricities ($e=\sqrt{1-{\left(b/a\right)}^{2}}$) values depends on aspect ratio (b/a) correspond to, semi minor axis /semi major axis [33]. The approximate values of Mathieu function ($q_{11}^{e,o})$ can be determined by following set of equations discussed in [32]. It can be seen that dual resonant modes of an elliptic shape structure can be controlled by varying the aspect ratio (b/a) that can be optimized to exhibit larger bandwidth.

In Yagi antennas, induced current exhibits progressive phase shift on the rest of parasitic elements which establishes an array structure suitable for supporting travelling wave. For the circuits printed over a dielectric material, it is essential to present a smooth transitions of induced electric field from the first directive element to the last directive element over the complete band of interest. However, abrupt discontinuity because of sharp edges of a rectangular shape geometry, the electric field lines bends at the edges which results in fringing electric fields. Therefore, to avoid sudden discontinuities along the whole structure, we selected elliptical shape geometry for our design. The results revealed by multiple electromagnetic simulations, we found the antenna’s elements with an elliptical shape geometry improves the overall performance especially the bandwidth thus support the travelling wave phenomenon better than its counterpart structure.

The general design principles and analytical investigations related to printed Yagi antenna has been extensively discussed in the literature [34,35,36]. However, it is important to summarize key salient guidelines required to understand the design and physical operational mechanism of the antenna. The printed Yagi antenna requires a balanced feed to the double sided printed dipole elements. Therefore, a 50 Ω microstrip to a parallel strip line transition is required. Typically, printed Yagi structure exhibits, low input impedance and lower front to back ratio. These parameters are significantly controlled, respectively (i) the spacing between the active dipole arms and the ground plane called reflector and (ii) the size (typically the length) of the reflector. By increasing the size of reflector, the backward gain reduces resulting in an improved front-to-back ratio. Similarly, the input impedance increase by increasing the separation between the dipoles and the reflector. However, a significant drop in forward gain is observed for a reflector spacing more than 0.3λ.

In travelling wave antennas, gain enhancement is the function of longitudinal dimensions thus proportional to the larger dimensions (length) of the antenna. In Yagi antennas gain increases by increasing the number of directive elements. However, practically, there is a limit beyond that additional directive element enhances very little to the gain value because of gradual reduction in induced electric field on the farthest directive elements. For our gain requirements (8 dBi~9 dBi), we found maximum four equally spaced directive elements are suitable resulting in overall antenna length to be 140 mm including the ground plane (reflector). As the guided wavelength is smaller than free space wavelength by a factor of square root of the relative permittivity of the substrate material, the major design parameters such as dipole arm lengths, length and spacing between the directive elements are normalized accordingly. Considering the same number of directive elements, the initial length and width of the antenna without dielectric material was calculated to be 285 mm and 85 mm, respectively. It can be seen that the use of substrate material reduces the length by more than half.

In summary, typical performance characteristics parameters of a printed Yagi antenna are affected by (i) reflector and feeder geometry (ii) spacing between reflector and the feeder and (iii) the total numbers of directive elements. Therefore, to achieve required performance it is essential to optimize the geometrical parameters of these structural variables.

## 3. Proposed Yagi Antenna Design

The location and geometric parameter (length and width) of the directive elements is very important in designing a Yagi antenna element. The input impedance of the antenna element is mainly affected by the first parasitic directive element. We have exploited this fact to design a wideband Yagi antenna. At first, we designed a Yagi antenna using conventional design approach, where first parasitic directive element is placed at λ_g_/4 from the active dipoles, towards end-fire direction as shown in Figure 1. The layout diagram comprising top and bottom planes of the antenna structure are shown in Figure 1a,b, respectively. The design procedure is initiated by the elliptically shaped driven elements of λ_g_/4 length printed on both metallization planes of the substrate. The truncated ground plane layer acts as a reflector for the Yagi structure. The antenna is fed by a 50 Ω microstrip line printed on the ground plane. The plated-via holes are placed on the two sides (left and right) of the printed circuit board (PCB). These holes are capable of reducing the side-lobe levels (SLL) by concentrating the radiated fields toward the end-fire direction [37]. The antenna is designed on a FR4 substrate with relative permittivity, dissipation factor and thickness of 4.4, 0.025 and 1.6 mm, respectively. By placing directive elements toward the end-fire direction of the antenna, the electromagnetic energy can be concentrated and a broad bandwidth matched to 50 Ω input impedance can be achieved. Therefore, to achieve wide bandwidth, we introduced an additional parasitic directive element between the active dipole arms and first directive element called tightly coupled directive element. The first directive element is wider compared to the others, and it is placed in close proximity to the driven elements. The top and bottom plane drawings of the proposed design having additional directive element is shown in Figure 1c and d, respectively. This placement is quite uncommon in conventional printed Yagi-antenna design. It is worth mentioning that the directive elements are placed on the top and bottom planes of the antenna. To achieve efficient electric-field coupling among the directive elements, the bottom plane directive elements are placed with some offset with respect to the top plane directive elements. The purpose is to enhance the effective width toward the direction of propagation. Critical antenna design parameters, such as the first directive-element spacing, length, width, the other directive-elements dimensions (lengths and widths), and the spacing between directive elements, are optimized using the full-wave electromagnetic simulation program CST Microwave Studio 2019 [38]. The final antenna dimensions used in the fabrication are presented in Table 2.

## 4. Parametric Study and Analysis of Geometrical Antenna Parameters

In a standard travelling-wave printed Yagi antenna, the effect of various geometrical antenna parameters such as (i) driven elements’ width and length (ii) reflector’s spacing, width and length (iii) different or equal length directive elements (iv) number of directive elements and (v) different or equal width directive elements, on critical performance parameter, has been extensively studied in the literature [34,35,36]. In our design, at first step we optimized each geometric parameter of the standard printed Yagi’s elements to their values resulting the optimum performance of the antenna. For simplicity, we do not report parametric studies of these well-known geometric variables and their effect on antenna’s performance. Instead, in this section, we analyze the effect of geometric parameters of the proposed tightly coupled directive element on the gain and reflection coefficient characteristics of the antenna. By multiple full-wave simulation analysis, we studied the effect of length, width and separations (r_nf_, r_nw_, d_4_ and d_5_ in Table 2) of the tightly coupled directive element on the gain and bandwidth of the proposed antenna. The length and width of the elliptic shape directive element are presented by their major and minor axis diameters, r_md_ and r_nw_, respectively. At first, by varying major and minor axis diameters of the said directive element, the variation inreflection coefficient and gain values are studied as shown in Figure 2. The major and the minor axis diameters’ values are varied over a range 8.2–32.8 mm and 4.4–8.8 mm, respectively. In Figure 2a, the major axis diameter is changed while keeping the minor axis diameter constant to a value of 4.4 mm and so on to 6.6 mm and 8.8 mm as shown in Figure 2b,c, respectively. It can be seen that the printed Yagi antenna experiences a significant change in bandwidth and gain by varying major axis diameter more than the minor axis diameter values. By analyzing these results, the optimum value of major and minor axis diameters are acquired to be 24.6 mm and 6.6 mm, respectively. In the next simulation setup, the position of the proposed directive element is optimized. In Figure 3, it can be seen that translating the said directive element away from the active dipoles critically effects the reflection coefficient thus the gain values over the proposed spectrum. Conversely, while the directive element is placed closer to the dipole arms, the proposed antenna still exhibited a wide bandwidth performance.

Finally, the performance of the proposed antenna element having a wide tightly coupled directive element, is studied and compared with the antenna without wide directicetive element. The major performance parameters such as reflection coefficient, realised gain and input impedance of both antenna elements are calculted and plotted. The simulated results of reflection cofficient (S_11_) and realised gain are shown in Figure 4a,b, respectively. The antenna without wide directive element presents lower bandwidth as compared to the proposed design. The former antenna element exhbits reflection coefficient values (parameter S_11)_ less than −10 dB in the frequency range 2.5–3.2 GHz, which correspond to a 28% fractional bandwidth value. On the other hand, the proposed antenna presents S_11_ less than −10 dB in the frequency range 2.6–4.3 GHz correspond to a 49.3% fractional bandwidth value. Similarly, the presence of tightly coupled directive element enhances realized gain value by more than 1.5 dB. The maximum realized gain values of the Yagi without and with wide directive element are found 7.1 dBi and 8.6 dBi, respectively. To illustrate the effect on input impedance of planar Yagi antenna because of additional directive element, Z-parameter of both designs are calculated and compared shown in Figure 5. The calculated Z_11_ (real) and Z_11_ (imaginary) are shown in Figure 5a,b, respectively. It can be seen that for the antenna with wide directive element, Z_11_ (real) values are close to 50 over a wider bandwidth as compared to the antenna without wide directive element. Similarly, the antenna with wide directive element, Z_11_ (imaginary) values are close to “zero” over a wider bandwidth as compared to the antenna without wide directive element. It can be concluded that presence of additional parasitic directive element in close proximity of the active dipoles does not increase the overall antenna dimensions of a planar Yagi antenna. However, it significantly improves the bandwidth and gain parameters simultaneously.

## 5. Antenna Fabrication and Measurements

The designed antenna was fabricated using a conventional chemical-based fabrication process. A moisture and dust green-solder mask was used to protect the metalized layers from oxidation. The spacing used between the plated-through holes of 1-mm in diameter was 1.6 mm. These holes were fabricated using a conventional drilling and plating process. The overall dimensions of the proposed antenna structure are 60 × 140 mm^2^. A photograph of the fabricated Yagi antenna is shown in Figure 6. The measured and simulated reflection coefficient results (parameter S_11_) are presented in Figure 7. The reflection coefficient values (parameter S_11_) of the proposed antenna are less than −10 dB in the frequency range 2.6–4.3 GHz, which is equivalent to a fractional bandwidth of 49.3%.

The simulated and measured E-plane (xy-plane) and H-Plane (yz-plane) radiation patterns at 3 GHz, 3.5 GHz, and 4.0 GHz are presented in Figure 8a–c, respectively. Good agreement is observed between measured and simulation results. The proposed antenna exhibits excellent radiation characteristics with the maximum value of the side-lobe level (SLL) being less than −10 dB compared to the main lobe. Other radiation parameters, such as the half-power beam width (HPBW), cross-polarization, realized gain, and side-lobe levels for the E-plane and H-plane radiation patterns at different frequencies are summarized in Table 3.

The 3D-radiation pattern graphs at 2.75 GHz and 3.5 GHz are presented in Figure 9a,b, respectively. At higher frequencies, the antenna beams become more directive and concentrates toward the end-fire direction. A snapshot photograph presenting the electric field distribution over the top plane of the antenna’s surfaces with and without plating via-holes is shown in Figure 10a,b, respectively. It can be observed that via-holes concentrate maximum energy toward the end-fire direction. On the other hand, in the antenna without via-holes, a strong electric field near the edges is observed. The antenna gain was measured using standard gain horn-antenna measurements. Initially, the free-space transmission coefficient (parameter S_21_) was measured using (i) a broadband horn antenna (LB-10180) from the A-info company and (ii) a standard-gain horn (SGH) antenna from the NSI company. Next, the standard-gain horn antenna was replaced by the antenna under test (printed Yagi), and the S_21_ measurements were recorded. The gain values were calculated by consulting the SGH data-sheet and comparing both readings. A comparison between the simulated and measured gain performance is shown in Figure 11a.

Both simulated and measured results are in good agreement with a difference not exceeding ± 0.8 dB. Commonly, such difference may occur due to connector’s insertion loss, fabrication tolerances, radiation pattern measurements performed in non-anechoic chamber environment, difference in substrate material definition used in simulation and real fabrication especially the dissipation factor (tanδ) value. From the parametric analysis, it can be discovered that antenna’s performance parameters especially the reflection coefficient and the gain values are significantly affected by the varying structural parameters in the order of few millimeters. The simulated antenna’s directivity and gain at 3.5 GHz are found to be 9.93 dBi and 8.31 dBi, respectively. This difference might be due to the fabrication tolerances of the antenna manufacturing process. Secondly, we considered that the substrate used in fabrication may have exhibited more dissipation loss than that one used in our simulation setup. 

It is important to calculate the antenna’s total efficiency as following [39]:(2)$$\in_{T}=\;\in_{R}\times \in_{lm}$$
where, $\in_{R}$ and $\in_{lm}$, correspond to radiation efficiency and mismatch loss, respectively. Congruently, total efficiency is defined as the ratio of the total power radiated by the antenna to the total power input to the antenna [40]
(3)$$\in_{T}=\;P_{rad}/P_{input}$$
however, the radiation efficiency comprises the conduction and dielectric losses and it can be calculated by the total power radiated by the antenna to the total power accepted by the antenna [40],
(4)$$\in_{R}=P_{rad}/P_{accept}$$
likewise, mismatch loss is [40]
(5)$$\in_{lm}=1-{\left|Г\right|}^{2}$$
where, Г correspond to voltage reflection coefficient at antenna’s input terminal. If there is no mismatch at the input terminal of the antenna the radiation efficiency will correspond to the total efficiency.

Antenna’s radiation efficiency is very important figure of merit for its characterization. Many measurements methods such as pattern integration, directivity/gain, wheeler cap, sliding wall cavity and Q-factor has been developed to find antenna’s radiation efficiency. A comprehensive discussion on different efficiency measurement methods is found in [40]. Among several state of the art techniques, efficiency calculation based on directivity and gain values is found to be simple, accurate and fast [40]. We know antenna’s radiation efficiency is related to gain (G) and directivity (D) as following [34]:(6)η=G/D
and for a directional antenna the directivity is closely linked to the horizontal $\theta_{HP}$ and vertical $\varphi_{HP}$ beam-widths as following [41]:(7)$$D\approx \frac{4\pi }{\theta_{HP}\varphi_{HP}}$$
due to significant advancement in electromagnetic simulations tools, it is easy to determine antenna’s directivity using numerical simulations. Therefore, antenna’s radiation efficiency is calculated using measured gain and simulated directivity values as following [34]:(8)$$\%\in_{R}=10^{-\left[\frac{\left(D-G\right)}{10}\right]}\times 100$$
finally, having simulated and measured reflection coefficients values, the simulated and measured total efficiency is calculated by using (8) and (5). The graph presenting simulated and measured total efficiency is shown in Figure 11b. Due to higher dissipation factor of FR4, proposed antenna’s efficiency is worse at higher frequencies. The antenna performance for a wide-band signal transmission can be characterized by calculating the following fidelity factor [42]:(9)$$FF=max\left|\frac{\int_{-\infty }^{\infty }x_{1}\left(t\right)*x_{2}\left(t+t_{d}\right)dt}{\sqrt{\int_{-\infty }^{\infty }{\left|x_{1}\left(t\right)\right|}^{2}dt}\;\sqrt{\int_{-\infty }^{\infty }{\left|x_{2}\left(t\right)\right|}^{2}dt}}\right|$$
where, $x_{1}\left(t\right)$ and $x_{2}\left(t\right)$ are transmitted and received signals, respectively. The antenna’s response for a wideband excitation signal was recorded in the CST simulation program using an *x*-oriented Electric Far-field probe, which was placed at 500 cm toward the radiation direction (along *y*-axis). Using (1), the pulse fidelity factor was calculated to be 0.97. It can be concluded that the proposed printed Yagi antenna is capable of radiating broadband signals with low dispersion.

## 6. Conclusions

In this paper, the concept of bandwidth enhancement of a conventional printed Yagi bi-layer antenna operating at the S-band was presented. Both metalized layers at the top and bottom antenna planes comprise ground plane, microstrip feedline, driven elements, and directive elements. The bottom plane directive elements are placed with some offset toward the direction of propagation to enhance the overall effective width. The aim was to simultaneously maximize gain and bandwidth. To achieve broad bandwidth matched to 50 Ω input impedance, elliptically-shaped directive elements were introduced in close proximity of the active dipole elements. Full-wave analysis of the tightly-coupled directive elements was performed by conducting multiple electromagnetic simulations using CST Microwave Studio. The performance of proposed antenna element having wide directive element is compared with the typical design without the wide directive element. Compared to the traditional Yagi antenna, the proposed antenna outperformed by presenting simultaneously, larger bandwidth and higher gain values. The designed antenna exhibited a 49.3% fractional bandwidth operating in the frequency range 2.6−4.3 GHz. Plated via-holes were introduced near the left and right boundaries of the PCB board to concentrate the electromagnetic energy toward the main direction of the electric-field flow. The proposed antenna exhibits excellent radiation performance with an average gain of more than 7 dBi and maximum side-lobe levels of less than −10 dB. Due to the high-fidelity factor value, the proposed antenna can be used in broadband RF applications.

## Figures and Tables

**Figure 1 micromachines-11-00975-f001:**
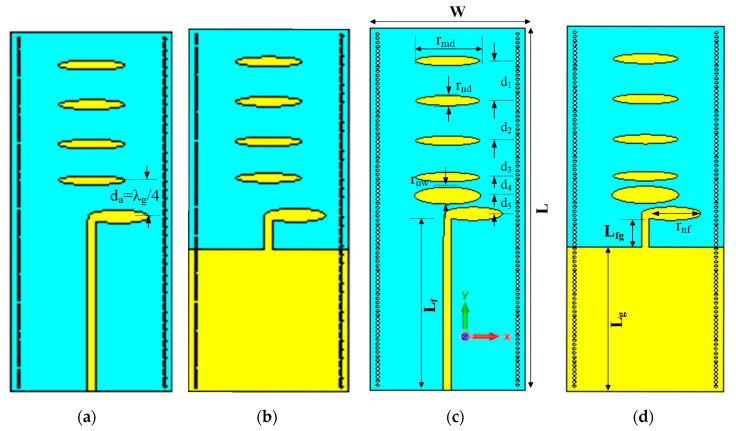
Layout diagram of the Yagi antenna (**a**,**b**), with first directive element at λ_g_/4 and geometric parameters of the proposed antenna design (**c**,**d**), with tightly coupled directive element.

**Figure 2 micromachines-11-00975-f002:**
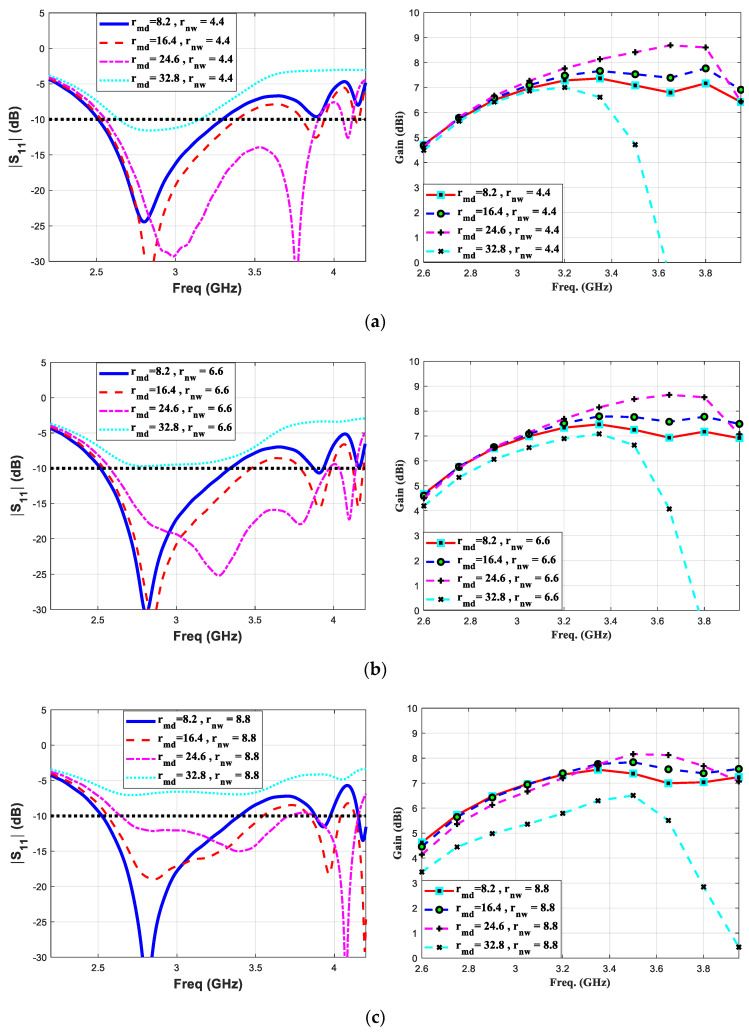
Parametric study of proposed antenna structural parameters based on reflection coefficient and gain performance analysis with (**a**) r_nw =_ 4.4 mm, (**b**) r_nw_ = 6.6 mm and (**c**) r_nw_ = 8.8 mm.

**Figure 3 micromachines-11-00975-f003:**
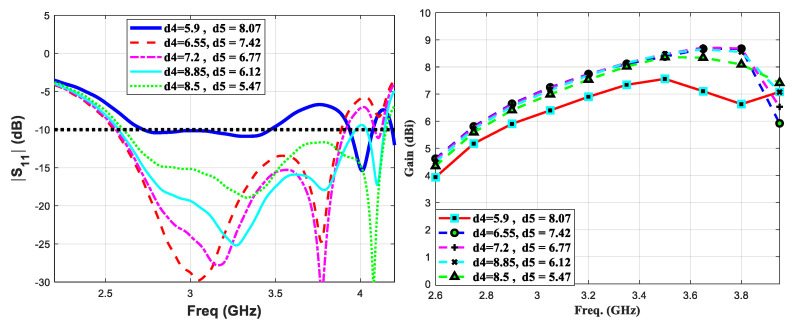
Parametric study of the position of the proposed directive element based on reflection coefficient and gain performance analysis.

**Figure 4 micromachines-11-00975-f004:**
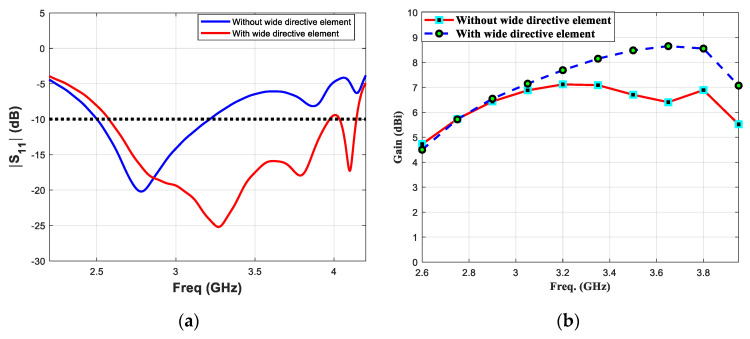
Comparison of (**a**) reflection coefficient (S_11_) and (**b**) realized gain parameters of the proposed antenna element, without and with tightly coupled wide directive element.

**Figure 5 micromachines-11-00975-f005:**
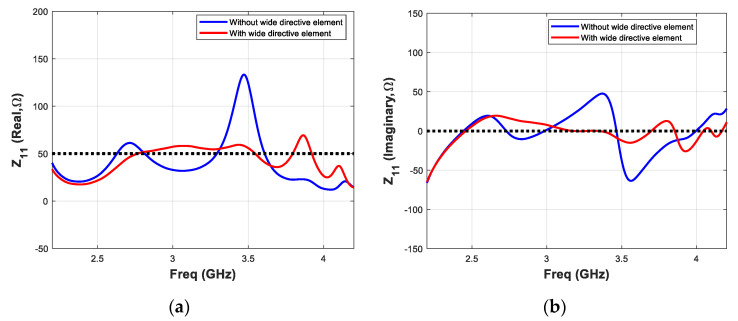
Comparison of (**a**) input impedance (real) and (**b**) input impedance (imaginary) parameters of the proposed antenna element, without and with tightly coupled wide directive element.

**Figure 6 micromachines-11-00975-f006:**
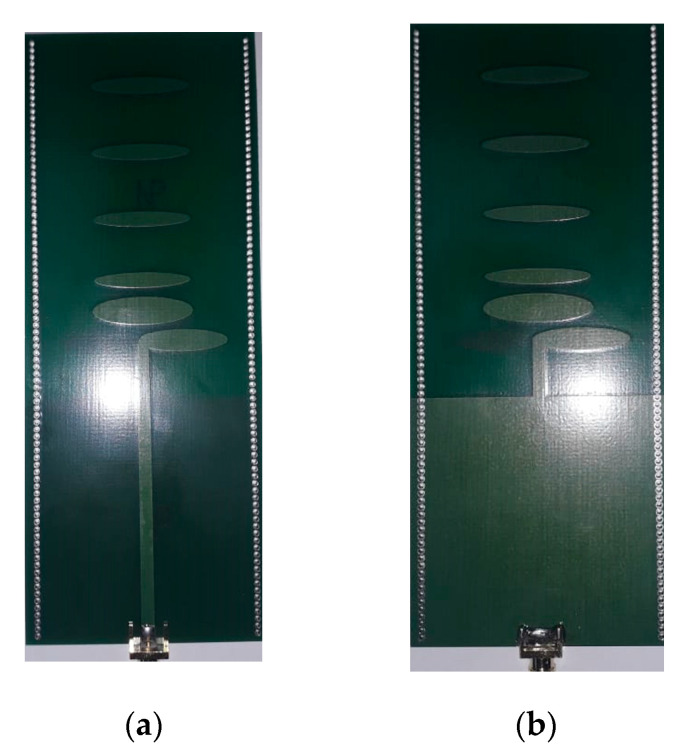
Photograph of the fabricated proposed antenna (**a**) front view (**b**) back view.

**Figure 7 micromachines-11-00975-f007:**
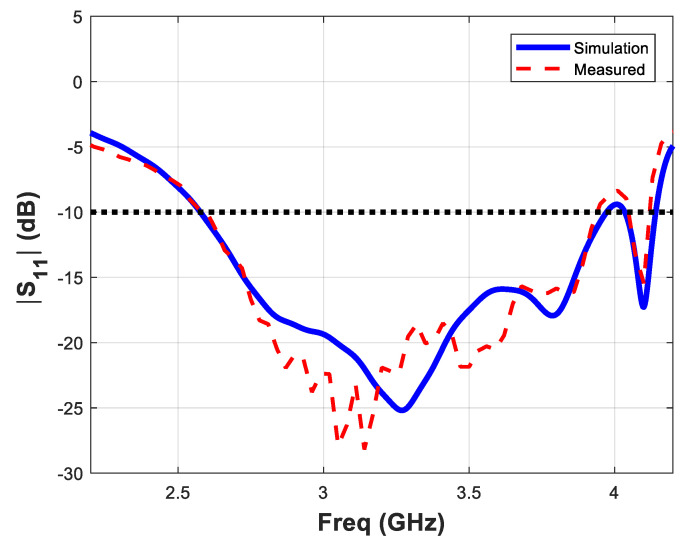
Measured and simulated reflection coefficient response of the printed Yagi antenna.

**Figure 8 micromachines-11-00975-f008:**
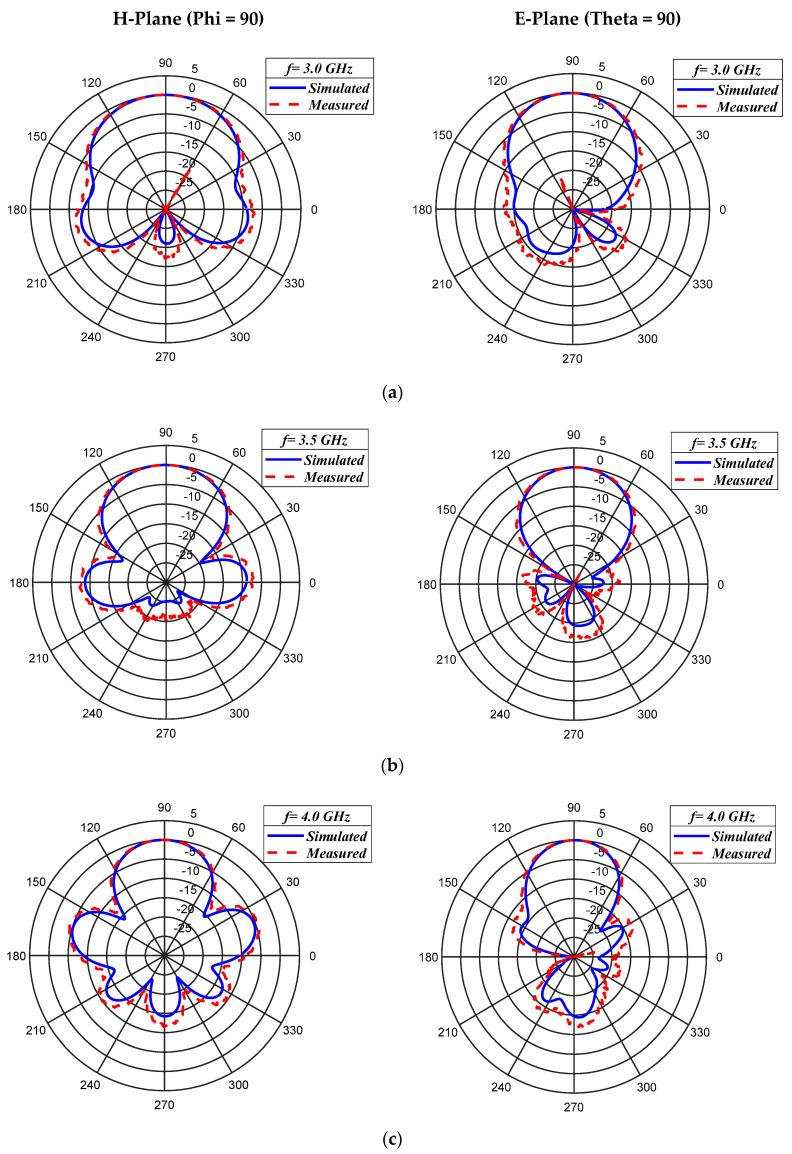
Simulated and measured normalized radiation patterns in E-Plane and H-plane of the proposed antenna, (**a**) *f* = 3 GHz, (**b**) *f* = 3.5 GHz and (**c**) *f* = 4.0 GHz.

**Figure 9 micromachines-11-00975-f009:**
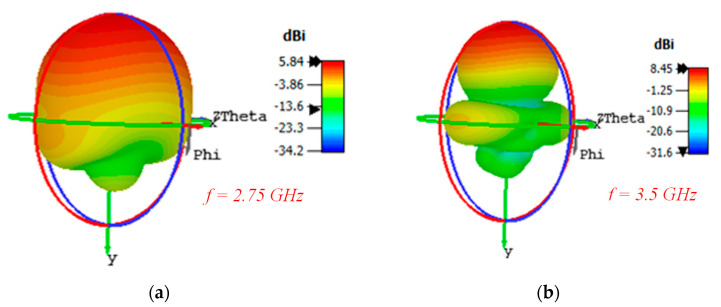
Simulated 3D radiation pattern patterns of the proposed antenna (**a**) *f =* 2.75 GHz and (**b***) f =* 3.5 GHz.

**Figure 10 micromachines-11-00975-f010:**
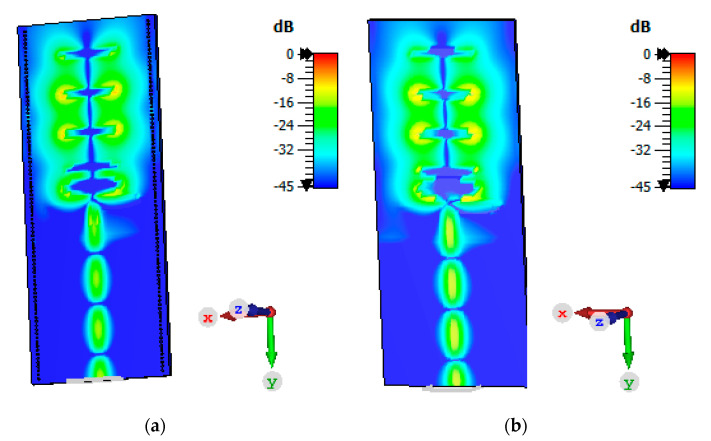
Snapshot photograph presenting the electric field distribution (**a**) with via-holes and (**b**) without via-holes.

**Figure 11 micromachines-11-00975-f011:**
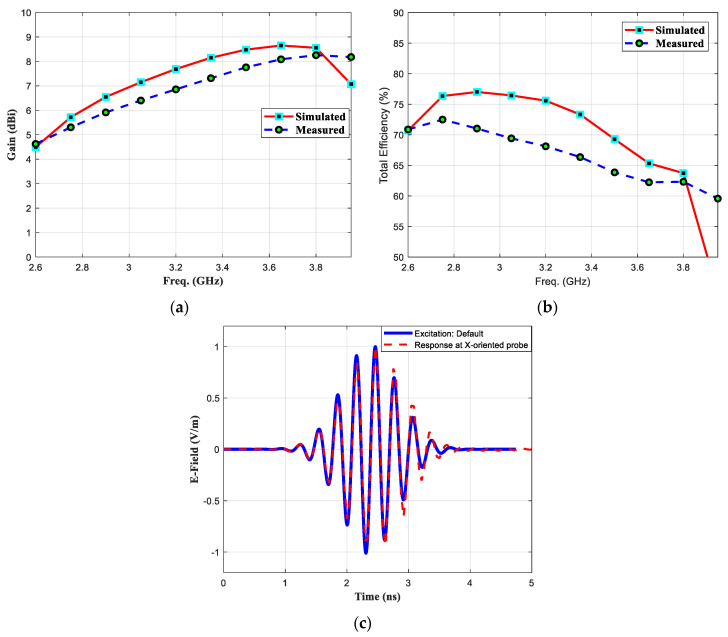
Proposed Antenna’s (**a**) Gain vs. frequency (**b**) % efficiency vs. frequency and (**c**) time-domain excitation response received by the x-oriented probe.

**Table 1 micromachines-11-00975-t001:** Comparison of wideband printed Yagi Antenna with proposed work.

Ref.	Center Freq. (GHz)	BW (%)	Gain (Max)	Dimensions (L × W)
[28]	2.72	42	4	0.39λ_0_ × 0.47λ_0_
[29]	8.74	3	8.34	1.15λ_0_ × 1.15λ_0_
[30]	2.44	3.2	5	0.84λ_0_ × 0.58λ_0_
[31]	2.31	63.54	5.8	0.41 λ_0_ × 0.41λ_0_
Proposed	3.6	49.3	8.7	1.68λ_0_ × 0.72λ_0_

**Table 2 micromachines-11-00975-t002:** Optimized parameters of the proposed antenna in mm.

L	W	L_f_	L_fg_	L_g_	r_nf_	r_nw_
140	60	67	12.9	55.5	22.6	6.6
**r_md_**	**r_nd_**	**d_1_**	**d_2_**	**d_3_**	**d_4_**	**d_5_**
24.6	4.3	15.47	15.27	13.95	7.2	6.77

**Table 3 micromachines-11-00975-t003:** Summary of the proposed antenna radiation characteristics.

Frequency (GHz)	Gain (dBi)	HPBW (E-plane)	HPBW (H-plane)	SLL (E-plane)	SLL (H-plane)
2.9	6.5	63	88	−15	−13
3.5	8.4	53	65	−17	−12
3.8	8.2	45	53	−14	−10

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
