# Peer review of "Design and Development of a Wideband Planar Yagi Antenna Using Tightly Coupled Directive Element"

_micromachines, 2020, doi:10.3390/mi11110975_

Round 1
Reviewer 1 Report
Gain and bandwidth improvement using the proposed design is of interest. The optimized design is achieved computationally. If you can add more physical/theoretical insights or guidelines into the design optimization that would would be valuable.
Author Response
Gain and bandwidth improvement using the proposed design is of interest. The optimized design is achieved computationally. If you can add more physical/theoretical insights or guidelines into the design optimization that would be valuable.
We would like to thank reviewer’s valuable comment. We have included a new heading named “Printed Yagi Antenna Design Principle”. In this section, we have elaborated the key guidelines required to understand the design and physical operations of the antenna. Please see highlighted on page 3 lines 89- 127.
Reviewer 2 Report
In this paper, the authors proposed a Design and Development of a Wideband Planar Yagi Antenna Using Tightly Coupled Directive Element. The system is operating S-band.
- What is the interest of using the S- band in modern wireless communication?
- What is the difference between the elliptically and the rectangular shape? No theory is provided.
- The circuit size is depending on the material. Therefore, the initial design should be provided for comparison.
- The references provided are not sufficient for the state of the art. Authors must expand their literature overview with the recent works.
- The manuscript is poorly written with a lot of typos
- The novelty of the paper is limited and questionable
- What is the novelty of the proposed design compared to the existing designs?
Author Response
In this paper, the authors proposed a Design and Development of a Wideband Planar Yagi Antenna Using Tightly Coupled Directive Element. The system is operating S-band.
1.What is the interest of using the S- band in modern wireless communication?
We would like to add that the mid-band 5G (5th generation communication systems) uses microwaves of 2.5-3.7 GHz, currently allowing speeds of 100-900 Mbit/s, with each cell tower providing service up to several miles in radius. For further details, we refer the reviewers to the following references that are addressing his comment:
- Christian (March 27, 2020). "What is 5G? The next-generation network explained". Digital Trends. Retrieved April 25, 2020.
- Hoffman, Chris (January 7, 2019). "What is 5G, and how fast will it be?". How-To Geek website. How-To Geek LLC. Archivedfrom the original on January 24, 2019. Retrieved January 23, 2019.
- Horwitz, Jeremy (December 10, 2019). "The definitive guide to 5G low, mid, and high band speeds". VentureBeat online magazine. Retrieved April 23, 2020.
- Davies, Darrell (May 20, 2019). "Small Cells – Big in 5G". Nokia. Retrieved August 29, 2020.
2. What is the difference between the elliptically and the rectangular shape? No theory is provided.
In Yagi antennas, induced current exhibits progressive phase shift on the rest of parasitic elements which establishes an array structure suitable for supporting travelling wave. For the circuits printed over a dielectric material, it is essential to present a smooth transitions of induced electric field from the first directive element to the last directive element over the complete band of interest. However, abrupt discontinuity because of sharp edges of a rectangular shape geometry, the electric field lines bends at the edges which results in fringing electric fields. Therefore, to avoid sudden discontinuities along the whole structure, we selected elliptical shape geometry for our design. The results revealed by multiple electromagnetic simulations, we found antenna elements with elliptical shape geometry improves the overall antenna’s performance especially the bandwidth thus support the travelling wave phenomenon better than its counterpart structure.
The manuscript is updated as well and changes are highlighted on Page 3 Line 101~110.
3. The circuit size is depending on the material. Therefore, the initial design should be provided for comparison.
In travelling wave antennas, gain enhancement is the function of longitudinal dimensions thus proportional to the larger dimensions (length) of the antenna. In Yagi antennas gain increases by increasing the number of directive elements. However, practically, there is a limit beyond that additional directive element enhances very little to the gain value because of gradual reduction in induced electric field on the farthest directive elements. For our gain requirements (8 dBi ~ 9 dBi), we found maximum four equally spaced directive elements are suitable resulting in overall antenna length to be 140 mm including the ground plane (reflector). As the guided wavelength is smaller than free space wavelength by a factor of square root of the relative permittivity of the substrate material, the major design parameters such as dipole arm lengths, length and spacing between the directive elements are normalized accordingly. Considering the same number of directive elements, the initial length and width of the antenna without dielectric material was calculated to be 285 mm and 85 mm, respectively. It can be seen that the use of substrate material reduces the length by more than half value.
The manuscript is updated as well and changes are highlighted on Page 3 Lines 111~123.
4.The references provided are not sufficient for the state of the art. Authors must expand their literature overview with the recent works.
We would like to thank reviewer for his valuable comment. We have updated introduction and expanded the literature review by introducing more references. In updated version, changes are highlighted on Page 2 Lines 58~72.
5.The manuscript is poorly written with a lot of typos.
The updated version has considered to address this weakness by the revision done by Co-authors.
6.The novelty of the paper is limited and questionable.
Thank you highlighting this important inadequacy. According to reviewer’s comments, introduction part is updated. Our work is compared with some of latest work summarized and Table 1.
7.What is the novelty of the proposed design compared to the existing designs?
According to author’s understanding, printed Yagi antenna with tightly coupled director element is a novel idea used for bandwidth enhancement. Moreover, this additional directive element contributes to the gain value, not less than a dB, without increasing the antenna’s length.
Reviewer 3 Report
1. It is recommended to add the following reference where the design of dual-band quasi-Yagi antenna operating at 1.8 and 2.5 GHz is presented:
- Z. Chen, M. Zeng, A.S. Andrenko, Y. Xu, and H.-Z. Tan. A dual-band high-gain quasi-Yagi antenna with split-ring resonators for radio frequency energy harvesting. Microwave Opt. Technology Lett., 2019, 61, pp. 2174–2181. https://doi.org/10.1002/mop.31872
2. The parametric study and analysis of the effect of geometrical antenna parameters on the reflection coefficient and wideband performance are missing. Table 2 shows the optimized antenna parameters. It is recommended to add the numerical results of S-11 parameter versus the parameters d4 and d5.
3. In Figure 8, it is hard to see that "It can be observed that via-holes concentrate maximum energy toward the end-fire direction. On the other hand, in the antenna without via-holes, a strong electric field near the edges is observed" as claimed on Page 8. Please present additional E-field distributions taken at different time instances.
4. What is the reason of a substantial difference (close to 1dB) between the calculated and measured gain presented in Figure 9a? It is recommended to measure and present the total antenna efficiency versus frequency.
Author Response
- It is recommended to add the following reference where the design of dual-band quasi-Yagi antenna operating at 1.8 and 2.5 GHz is presented:
- Chen, M. Zeng, A.S. Andrenko, Y. Xu, and H.-Z. Tan. A dual-band high-gain quasi-Yagi antenna with split-ring resonators for radio frequency energy harvesting. Microwave Opt. Technology Lett., 2019, 61, pp. 2174–2181. https://doi.org/10.1002/mop.31872
- Thank you for the comment. The reference is added at No. 25 and highlighted.
- The parametric study and analysis of the effect of geometrical antenna parameters on the reflection coefficient and wideband performance are missing.
We would like to thanks this valuable comment. Parametric study and analysis of important geometric structure is added in the revised version. In updated version, changes are highlighted on Page 5~7 Lines 166~204.
3. Table 2 shows the optimized antenna parameters. It is recommended to add the numerical results of S-11 parameter versus the parameters d4 and d5.
Thank you for your advice. Numerical results of S11 and gain are added and discussed on Page 6~7.
4. In Figure 8, it is hard to see that "It can be observed that via-holes concentrate maximum energy toward the end-fire direction. On the other hand, in the antenna without via-holes, a strong electric field near the edges is observed" as claimed on Page 8. Please present additional E-field distributions taken at different time instances.
Thank you for highlighting. The said figure is updated. It can be seen under Figure 10 Page 11 of the updated version.
5.What is the reason of a substantial difference (close to 1dB) between the calculated and measured gain presented in Figure 9a?
We are very thankful for pointing out the difference between the calculated and measured gain values. Commonly, such difference may occur due to connector’s insertion loss, fabrication tolerances, radiation pattern measurements performed in non-anechoic chamber environment, difference in substrate material definition used in simulation and real fabrication especially the dissipation factor (tanδ) value. There are variety of commercial available FR4 substrate materials with different dissipation factor values. We consider that the substrate used in fabrication have exhibited more dissipation lessee than that one used in our simulation setup. The simulated antenna’s directivity and gain at 3.5 GHz are found to be 9.93 dBi and 8.31 dBi, respectively. This significant difference is because of higher loss tangent value of the FR4 material causing significant dissipation of the waves inside the substrate.
The above discussion is also added to the current version can be seen on Page 11 Line 292~301.
6. It is recommended to measure and present the total antenna efficiency versus frequency.
The proposed antenna efficiency is calculated using measured gain values and added to revised manuscript under Figure 11.
Round 2
Reviewer 2 Report
The authors have addressed all my comments.
Author Response
Dear Sir;
Thank you for your update.
There is no comment.
Reviewer 3 Report
The content of the manuscript in its current form has been improved.
I have one concern related to my last comment in the initial review.
"6. It is recommended to measure and present the total antenna efficiency versus frequency."
The authors have presented the "antenna's radiation efficiency from the measured gain values" in Fig. 11b. However, the total antenna efficiency (rather than radiation efficiency as shown in Fig. 11b) provides an ultimate benchmark on the antenna performance. Again, it is recommended to measure and present the total antenna efficiency versus frequency.
Author Response
I have one concern related to my last comment in the initial review.
"6. It is recommended to measure and present the total antenna efficiency versus frequency."
The authors have presented the "antenna's radiation efficiency from the measured gain values" in Fig. 11b. However, the total antenna efficiency (rather than radiation efficiency as shown in Fig. 11b) provides an ultimate benchmark on the antenna performance. Again, it is recommended to measure and present the total antenna efficiency versus frequency.
Thank you for the comment, in our updated version the antenna total efficiency is measured using directivity/gain method. In current version, Figure 11 presents the total measured efficiency instead of radiation efficiency.
Manuscript is updated and highlighted on page 12.